# Inverse-Folding Design of Yeast Telomerase RNA Increases Activity In Vitro

**DOI:** 10.3390/ncrna9050051

**Published:** 2023-08-28

**Authors:** Kevin J. Lebo, David C. Zappulla

**Affiliations:** 1Department of Biology, Johns Hopkins University, Baltimore, MD 21218, USA; 2Department of Biological Sciences, Lehigh University, Bethlehem, PA 18015, USA

**Keywords:** RNA, non-coding RNA, TLC1, telomerase, telomere, senescence, inverse design, RNA secondary structure, Mfold, TERT, Est2

## Abstract

*Saccharomyces cerevisiae* telomerase RNA, TLC1, is an 1157 nt non-coding RNA that functions as both a template for DNA synthesis and a flexible scaffold for telomerase RNP holoenzyme protein subunits. The tractable budding yeast system has provided landmark discoveries about telomere biology in vivo, but yeast telomerase research has been hampered by the fact that the large TLC1 RNA subunit does not support robust telomerase activity in vitro. In contrast, 155–500 nt miniaturized TLC1 alleles comprising the catalytic core domain and lacking the RNA’s long arms do reconstitute robust activity. We hypothesized that full-length TLC1 is prone to misfolding in vitro. To create a full-length yeast telomerase RNA, predicted to fold into its biologically relevant structure, we took an inverse RNA-folding approach, changing 59 nucleotides predicted to increase the energetic favorability of folding into the modeled native structure based on the p-num feature of Mfold software. The sequence changes lowered the predicted ∆G of this “determined-arm” allele, DA-TLC1, by 61 kcal/mol (−19%) compared to wild-type. We tested DA-TLC1 for reconstituted activity and found it to be ~5-fold more robust than wild-type TLC1, suggesting that the inverse-folding design indeed improved folding in vitro into a catalytically active conformation. We also tested if DA-TLC1 functions in vivo, discovering that it complements a *tlc1*∆ strain, allowing cells to avoid senescence and maintain telomeres of nearly wild-type length. However, all inverse-designed RNAs that we tested had reduced abundance in vivo. In particular, inverse-designing nearly all of the Ku arm caused a profound reduction in telomerase RNA abundance in the cell and very short telomeres. Overall, these results show that the inverse design of *S. cerevisiae* telomerase RNA increases activity in vitro, while reducing abundance in vivo. This study provides a biochemically and biologically tested approach to inverse-design RNAs using Mfold that could be useful for controlling RNA structure in basic research and biomedicine.

## 1. Introduction

Telomeres are the protein-bound repeated segments of DNA at the ends of linear chromosomes that help to define the natural DNA ends. Because canonical DNA polymerases are unable to fully replicate the ends of the chromosome, telomeres shorten with each successive generation [1]. If left unchecked, this will result in a G2/M cell cycle arrest known as senescence [2,3]. Many eukaryotes counteract this “end-replication problem” with a ribonucleoprotein complex known as telomerase [4]. Telomerase is minimally composed of a telomerase reverse transcriptase (TERT) protein and a telomerase RNA component [5,6]. TERT uses a portion of the RNA as a template for reverse transcription to lengthen the telomeric DNA [7]. In addition, several accessory protein subunits are required for telomere maintenance in vivo [8].

In the budding yeast *Saccharomyces cerevisiae*, the 1157 nt telomerase RNA is TLC1, which adopts an overall Y-shaped secondary structure with three long arms radiating out from the central catalytic core [9,10]. This central core secondary structure conforms to the conserved telomerase RNA core consensus model, containing the template and template-boundary element, a pseudoknot with catalytically important base triples, and a core-enclosing helix, all coordinated through an area of required connectivity [11,12,13]. Yeast TERT, Est2 (Ever-Shorter Telomeres 2), binds to this central core and functions with it to reconstitute core-enzyme activity [13,14,15,16].

In addition to providing the template for telomere synthesis by reverse transcription, TLC1 acts as a flexible scaffold to tether telomerase accessory proteins to the holoenzyme [10,17,18,19,20]. The accessory proteins, including the essential Est1 protein and the important Ku heterodimer and Sm_7_ complex, each bind to the end of a different long RNA arm [9,10,21,22,23]. Each of these accessory-protein binding sites can be relocated on the RNA with the retention of function; i.e., these subunits are flexibly scaffolded [10,19,20]. The bulk of the long RNA arms between the core and the protein-binding sites can be deleted without eliminating telomerase function in vivo, although RNA domains near the protein-binding sites may still have roles in telomerase function beyond protein tethering [14,24]. Overall, TLC1 provides a flexible scaffold that tethers holoenzyme protein subunits and RNA domains together without conferring specific orientations for function.

Although telomerase accessory protein subunits are required for telomere-length homeostasis in vivo, the RNA and TERT core enzyme components are minimally sufficient to reconstitute basic enzymatic activity in vitro [5,6,25]. In *S. cerevisiae*, this minimal telomerase RNP requires the TLC1 RNA and Est2 protein. However, full-length TLC1 only shows robust in vitro function when it has been co-immunopurified from yeast cells [14,23,25]. When yeast telomerase is reconstituted in vitro in a rabbit reticulocyte lysate transcription–translation system, wild-type TLC1 is quasi-nonfunctional, displaying only traces of telomerase activity [14,26,27].

To allow for in vitro telomerase studies, several mutant variants of TLC1 have been developed. These include a series of miniaturized telomerase RNAs, which involve extensive deletions of the TLC1 arms to make “Mini-T” [14] and “Micro-T” variants [11,16], or telomerase RNAs with significant structural modifications in the arms, such as triple-stiff arm TLC1 (TSA-T) [26]. Micro-T RNAs lack the essential protein-binding arms, and would therefore be nonfunctional in vivo [11,16]. Mini-T and TSA-T are able to support telomere maintenance in cells, and therefore can be used to study the RNA both in vivo and in vitro [14,26]. However, these TLC1 variants are highly mutated; Mini-T(500) has 657 nucleotides deleted from the three RNA arms, while Triple-Stiff Arm TLC1 has 201 nts deleted from the arms, and several other nucleotide substitutions. Furthermore, both Mini-T and TSA-T have dramatic structural changes from wild-type TLC1. The truncated arms of Mini-T bring the accessory proteins in closer to the central catalytic core, while the stiffened arms in TSA-T hold the accessory proteins away from the core on dsRNA struts. These mutations limit the amount of information that Mini-T and TSA-T can provide about the relationships between structure and function in yeast telomerase RNA. Therefore, we sought to engineer a version of TLC1 with fewer mutations that would be highly functional both in vivo and in vitro.

To design a full-length TLC1 RNA with sequence changes that promote folding into its active conformation (i.e., “inverse RNA folding” [28,29,30]), we employed the Mfold RNA secondary structure prediction software [31,32]. Mfold can model both the secondary structure and calculate the predicted ∆G of folded conformations. In addition, a feature of Mfold provides information about how “well-determined” the RNA structure is, based on a p-num heuristic descriptor [31,33,34]. p-num describes the propensity of a particular nucleotide to make alternative base pairings in iterations of Mfold folding simulations [34]. A high p-num value indicates that a nucleotide is predicted to pair with many other nucleotides, while a low p-num value reflects few energetically favorable pairing interactions with other nucleotides. Mfold uses a ROYGBIV spectrum scale to annotate the p-num values of each nucleotide, with red representing nucleotides with a low p-num value, and purple or black representing nucleotides with a high value. Red nucleotides in Mfold structures have a high probability of folding as drawn by Mfold in the lowest-free-energy structural conformation; domains with many red nucleotides are therefore more “well-determined,” with a relatively high confidence in the secondary structure model.

Here, we present “Determined Arm TLC1” (DA-TLC1), which retains the wild-type TLC1 length, but has an engineered RNA arm sequence that promotes these regions to be more determined to fold into their physiologically relevant structure. Compared to wild-type TLC1, the inverse-designed DA-TLC1 allele reconstitutes ~5-fold more enzymatic activity in vitro when paired with the Est2 protein, although it is not as robust as Mini-T or TSA-T RNAs. When expressed in vivo, DA-TLC1 provides basic functionality, allowing cells to avoid senescence and maintain shorter telomeres, with the alleles exhibiting reduced RNA abundance. Overall, DA-TLC1 represents a highly functional yeast telomerase RNA that will be useful for future studies of telomerase mechanism both in vivo and in vitro, and demonstrates that long noncoding RNAs with increased in vitro function can be inverse-designed using Mfold.

## 2. Results

### 2.1. Inverse Design of a TLC1 RNA That Folds More Readily into Its Active Biological Structure: Determined-Arm TLC1 (DA-TLC1)

To create a full-size TLC1 RNA that would fold into a functional conformation in vitro, we made a series of single-nucleotide substitutions in the RNA to optimize the lowest-free-energy folding of paired regions. The current model of TLC1 secondary structure, along with Mfold software predictions, were used to guide the inverse design of mutations that would create the most “well-determined” arms [34]. It is important to note that the Mfold secondary structure prediction for wild-type TLC1 (Figure 1A) varies from the phylogenetic models [9,10] in several locations. Firstly, Mfold does not predict pseudoknots (tertiary structure), although it predicts the formation of the two hairpins in the central core shown to be folding intermediates [35]. Secondly, some portions of the arms are predicted to fold differently; in particular, the region of the Ku arm distal to the core is very different between the phylogenetically derived [9,10] and Mfold model.

We performed ~36 rounds of manually testing mutations and their effects on Mfold p-num folding predictions to achieve an inverse-designed full-length telomerase RNA expected to fold more readily into its active secondary structure. “Determined Arm TLC1” (DA-TLC1) RNA has 59 substituted nucleotides across the 1157 nt RNA (Figure 1B; Appendix A). There are 11 nucleotide substitutions in the Ku arm, 33 substitutions in the Est1 arm, and 15 in the terminal arm (Appendix A). All the mutations were made within the core proximal regions of the arms, where no subunits have been shown to bind. These areas of TLC1 have been previously deleted in Mini-T [14] and stiffened in TSA-T [26] without abolishing telomerase holoenzyme function.

The lowest-free-energy Mfold secondary structure model for DA-TLC1 (Figure 1B) closely matches the secondary structure of the phylogenetically derived structure for wild-type TLC1, except in two major ways [9,10]. Firstly, the central catalytic core is predicted to fold differently from either the phylogenetically supported model [10] or Mfold of wild-type TLC1 (n.b., Mfold does not predict pseudoknots, and there is a pseudoknot in this region). Rather than forming the two-hairpin structure seen in the lowest-free-energy Mfold model of wild-type TLC1 (Figure 1A), the core of DA-TLC1 has only the 3′ hairpin, and the template is partially paired with nucleotides from the opposite side of the core. This is the same secondary structure predicted in the Mfold model for TSA-T [26], and could represent a biologically relevant folding conformation. Secondly, the distal region of the Ku arm in DA-TLC1 is predicted to fold similarly to the Mfold model for wild-type TLC1, rather than either of the phylogenetically predicted structures [9,10] (nts 176–409, Figure 1B). Because the two published phylogenetically supported structures differed in this region, we refrained from inverse-designing DA-TLC1 to conform to either one, rather leaving nucleotides 176–410 the same as wild type.

The Mfold model for DA-TLC1 also shows that each of its arms are energetically more “well determined” than those of wild-type TLC1 (Figure 1). More nucleotides in each of the three arms of DA-TLC1 are colored red on the p-num scale than in wild-type TLC1, indicating a higher degree of confidence in the indicated folded conformation, based on the program’s free-energy folding simulations (Figure 1). The arms of DA-TLC1 are therefore energetically more likely to fold into the predicted state than those in wild-type TLC1. Accordingly, DA-TLC1 is predicted to have 61 kcal/mol (19%) lower initial free energy of folding than TLC1 (−382 kcal/mol compared to −321 kcal/mol for wild type).

### 2.2. DA-TLC1 Retains Function In Vivo

We first tested if DA-TLC1 functions in vivo. DA-TLC1 RNA was expressed from a centromere-containing (*CEN*) plasmid, with the gene’s endogenous promoter and terminator, in a *tlc1*∆ *rad52*∆ strain. The deletion of *rad52*∆ prevents alternative recombination-based survivor pathways for telomere lengthening, and ensures that all cellular telomere lengthening is telomerase-dependent [36,37,38]. Cells were then restreaked 10 times, representing approximately 250 generations of growth.

DA-TLC1-expressing cells survived without senescing through the course of the ~20 days of cell passaging (Figure 2A). In contrast, cells lacking a functional telomerase RNA senesced by 125 generations of growth (i.e., within the first ~10 days). This demonstrates that the essential RNA domains in DA-TLC1, including the catalytic core and the Est1-binding arm, fold into functional structures in vivo.

To test quantitatively how well DA-TLC1 telomerase functions in vivo, we examined average telomere length and found it to be similar to wild-type cells. Telomeres were analyzed by Southern blotting genomic DNA from cells expressing DA-TLC1 for 250 generations. DA-TLC1 telomeres were slightly shorter than wild type, while telomeres maintained by TSA-T were moderately shortened, and those maintained by Mini-T were very short (Figure 2B). Thus, DA-TLC1 is able to maintain nearly wild-type telomeres in vivo, and better than the in vitro functional mutants TSA-T or Mini-T.

To measure DA-TLC1 RNA abundance compared to wild-type TLC1, we isolated total cellular RNA and analyzed it using Northern blotting, probing for the Mini-T sequence that is present in both TLC1 and DA-TLC1 transcripts. DA-TLC1 abundance in vivo was lower than wild-type (Figure 2C). However, DA-TLC1 abundance was somewhat higher than that of Mini-T RNA and similar to that of TSA-T [26] (Figure 2D). Taken together with the fact that DA-TLC1 cells have nearly wild-type telomere length, DA-TLC1 maintains telomeres effectively, especially considering its lower concentration in vivo.

### 2.3. DA-TLC1 Has Increased Activity In Vitro Compared to Wild-Type TLC1

The wild-type 1157 nt TLC1 RNA is barely able to support detectable telomerase enzyme function in an in vitro reconstituted telomerase activity assay [14,26]. We hypothesize that this is due to the misfolding of the central core and that the three long TLC1 arms interfere with the folding landscape of the catalytic core region. To test whether the more well-determined secondary structure in the arms of DA-TLC1 promotes telomerase activity in vitro, we performed RNP enzyme activity assays. DA-TLC1 was transcribed by T7 RNA polymerase alongside ProA-tagged yeast TERT, Est2, in a rabbit reticulocyte in vitro transcription and translation system [14]. Immunopurified telomerase RNP was incubated with a telomeric DNA primer and [α-^32^P]-dGTP, and products were separated using urea-polyacrylamide gel electrophoresis. DA-TLC1 successfully reconstituted telomerase function in vitro (Figure 3). The in vitro activity was reproducibly approximately 5-fold greater than wild-type TLC1, yet still substantially lower than Mini-T or TSA-T, each of which support very robust reconstituted activity. These data provide evidence that the well-determined inverse-designed arms of DA-TLC1 improve telomerase catalytic core function in vitro.

### 2.4. The Secondary Structure of the Ku Arm Affects Telomerase Function In Vivo and In Vitro

The Mfold secondary structure model for the Ku-binding arm of DA-TLC1 is similar to the Mfold-predicted folding of the wild-type TLC1 Ku arm (Figure 4A; Appendix A). However, this conformation differs from the models for the Ku arm secondary structure, which are constrained by covarying base pairs—i.e., phylogenetic support of folding that may not be, or be predicted to be, the lowest-free-energy structure [9,10]. Thus, we also inverse-designed the Ku arm with an alternative nucleotide sequence to Mfold into a phylogenetically supported structure [10] (Figure 4A, bottom). This “Determined Phylogenetic-structure Ku arm” (DPhyK) has an additional 51 point mutations compared to the DA-TLC1 Ku arm, extending distally out towards the Ku-binding site (Appendix A). We then compared a series of determined-arm variants for function in vivo and in vitro: DA-TLC1, DK-TLC1 (with the determined Ku arm from DA-TLC1 and wild-type Est1 and terminal arms), DET-TLC1 (with a wild-type Ku arm and the determined Est1 and terminal arms from DA-TLC1), DPhyK-DA-TLC1 (with the inverse-designed phylogenetically supported Ku arm conformation along with the determined Est1 and terminal arms found in DA-TLC1), and DPhyK-TLC1 (with the inverse-designed phylogenetically supported Ku arm conformation and the wild-type Est1 and terminal arms) (Figure 4 and Appendix A).

We first tested the Determined-Arm TLC1 RNA variants in vitro. Each of the various Ku-arm variants was able to reconstitute telomerase activity at levels near or slightly higher than DA-TLC1, although still lower than Mini-T (Figure 4B). These data suggest that driving the folding of the RNA arms by using Mfold to guide inverse-folding design supports increased telomerase core enzymatic activity. This is likely due to indirectly stabilizing the structure of the RNA core by limiting mispairing between nucleotides of the core and the Ku arm. We see this trend when we used a wild-type Ku arm with determined Est1 and terminal arms (Figure 4B, DET-TLC1), or even when only inverse-designing the Ku arm (Figure 4B, DK-TLC1 and DPhyK-TLC1).

After demonstrating that the catalytic cores of all the determined-arm variants were active in vitro, we next tested the in vivo function for these alleles, as performed above. All of the determined-arm variants complemented a *tlc1*∆ strain, without senescing over the course of the ~20 days of cell passaging (Figure 4C), indicating that the essential regions of TLC1, including the catalytic core and the Est1-binding arm, folded into functional structures in cells.

We next analyzed telomere lengths from cells expressing the determined-arm variants. Genomic DNA was isolated after 250 generations of growth and run on a Southern blot (Figure 4D). Like DA-TLC1, both DK-TLC1 and DET-TLC1 maintain near-wild-type telomeres, suggesting they form highly functional telomerase holoenzymes in vivo (Figure 4D, lanes 2−4). However, RNAs with the secondary-structure-driven Ku arm had very short telomeres, ~171–185 bp shorter than wild-type (lanes 5 and 6).

The telomere phenotypes may be the result of differences in RNA abundance. Telomerase RNA levels were assessed by Northern blots performed on total RNA isolated after 250 generations of growth. DK-TLC1 and DET-TLC1 had reduced RNA abundance compared to wild-type TLC1, with 25% and 40% of TLC1 levels, respectively, similar to DA-TLC1 (Figure 4E). This indicates that the mutations made in the determined arms result in lower telomerase RNA levels, despite the ability of the determined-arm TLC1 RNAs to maintain telomeres near wild-type length. On the other hand, constraining the Ku arm to the phylogenetically determined structure in DPhyK-DA-TLC1 and DPhyK-TLC1 results in extremely low RNA abundance, with 2–3% of TLC1 levels (Figure 4E). This reduction in telomerase RNA is likely responsible for the pronounced decrease in telomere length observed in these two extensively inverse-designed Ku-arm RNAs.

## 3. Discussion

The budding yeast *Saccharomyces cerevisiae* has been a powerful model organism for studying the genetics and molecular mechanisms of telomerase function, leading to landmark discoveries, including the first telomerase subunit gene and genetic discovery of the TERT catalytic protein subunit [6,8,39,40]. However, biochemical studies have been limited by the misfolding of full-length TLC1 in vitro, as evidenced by the failure to support telomerase enzyme function in an in vitro reconstituted telomerase assay. To bypass this problem, a series of miniaturized mutants of TLC1 (Mini-T alleles) were designed that restore in vitro activity [14].

Beyond Mini-T RNAs, yet further-reduced “Micro-T” RNAs have each of the three accessory-protein-binding arms entirely deleted, leaving just a ~170 nt central catalytic core [11,16]. However, although Micro-T is robustly functional in vitro, it would certainly be unable to function in vivo because it lacks essential accessory-protein-binding sites, such as the binding sites for Est1 and Pop1/6/7 proteins [22,24,27]. The slightly larger Mini-T RNAs are functional both in vitro and in vivo [14]. With 657 nts deleted from the RNA arms, Mini-T(500) is highly mutated in both sequence and structure, and the truncated arms draw the accessory-protein-binding sites in much closer to the catalytic core. Additionally, Mini-T RNA abundance is ~15% of wild type in vivo, and telomeres are maintained at a stable but much shorter length. Finally, TSA-T has the three accessory-protein-binding arms stiffened by the deletion of 201 nts and several nucleotide substitutions [26]. TSA-T is functional in vitro, similar to Mini-T and Micro-T. TSA-T is also highly functional in vivo, despite low RNA abundance. On a per-RNA basis, TSA-T actually maintains longer telomeres than wild-type TLC1, despite the fact that the stiffened arms of TSA-T significantly alter the overall architecture of the telomerase RNP. While the many heavily mutated TLC1 RNA mutants have helped make important advances in understanding the molecular mechanisms and organization of the yeast telomerase RNP, a minimally mutated RNA that folds into its native state in vitro would allow for the examination of the most biologically accurate RNA conformation and, ultimately, the assembly of the biologically relevant RNP complex. Being able to study such a minimally mutated TLC1 biochemically in vitro would advance understanding of both telomerase and the growing field of long noncoding RNAs (lncRNAs).

Paired regions in RNA structure are often replaceable with different base pairs without disturbing the structure or function of the RNA or RNP complex. In fact, this is a central tenet of determining RNA structure phylogenetically; nucleotides in a sequence alignment that are paired in the RNA transcript’s folded state may co-vary during evolution, allowing a researcher to infer that the nucleotides are paired. Thus, considering the extensive amount of natural covariation in RNA structures during evolution, it should not be expected that manually changing the identity of base pairs would necessarily disturb the structure or function at most positions of many RNAs. Furthermore, many unpaired nucleotides in loops and junctions often tolerate sequence substitutions; often, unpaired residues serve as flexible linkers between paired elements. These paradigms are particularly likely to hold true for telomerase RNA, given its rapid evolution in sequence and its demonstrated function in yeast as a flexible scaffold for holoenzyme subunits [10,11,19,20].

To design a minimally mutated wild-type-length TLC1 RNA that folds into its biologically relevant conformation and functions both in vivo and in vitro, we employed Mfold RNA secondary structure prediction software [31]. Using Mfold’s p-num heuristic descriptor function [31,34], which outputs colored nucleotides based on confidence in lowest-free-energy secondary structure modeling (i.e., “determinedness”), we introduced sequence substitutions in TLC1 to design a telomerase RNA with well-determined accessory-protein-binding arms. This was an iterative, manual process of introducing sequence changes and then testing if they had their intended effect on promoting the RNA fold into its physiologically relevant conformation based on Mfold p-num folding predictions. This inverse-folding design approach resulted in “Determined-Arm TLC1” (DA-TLC1), which is wild-type length (1157 nt) with 59 changed nucleotides (Figure 1 and Appendix A). The Mfold secondary structure model for DA-TLC1 is more similar than the wild-type TLC1 Mfold to the phylogenetically derived structure models for TLC1 [9,10], particularly at the base of the Terminal and Est1 arms (Figure 1; Appendix A), and also with a lower ∆G and increased p-num-based structural confidence (Figure 1 and Figure 4A).

Because DA-TLC1 is predicted to adopt a secondary structure similar to wild-type TLC1, it may prove to be a useful tool for studying the yeast telomerase flexible scaffold. Like wild-type TLC1, DA-TLC1 can prevent senescence and maintain nearly wild-type-length telomeres in vivo (Figure 2). Unlike TLC1, however, DA-TLC1 is substantially functional in vitro, extending a telomeric DNA primer in a reconstituted telomerase activity assay along with Est2 protein (Figure 3). The ability to transition between studying DA-TLC1 in vivo and in vitro could allow for further biochemical and biophysical analyses of yeast telomerase, advancing understanding of yeast telomerase RNP’s structure and function.

DA-TLC1 may also be superior to previously designed TLC1 variants with in vitro activity. While both Mini-T and TSA-T have higher in vitro function than DA-TLC1 (Figure 4), both of these RNAs have substantially shortened telomeres in vivo. Furthermore, with only 9.5% of the nucleotides mutated, DA-TLC1 is much more similar to wild-type TLC1 in sequence; Mini-T(500) has 657 nts deleted from the long RNA arms, while TSA-T has 223 nts deleted/mutated, resulting in long rigid dsRNA helices between the core and accessory proteins’ binding sites. In contrast, DA-TLC1 was designed to retain a wild-type-like secondary structure and is predicted to form an overall holoenzyme structure much more similar to that of TLC1. Changing bases within the paired segments of the RNA arms to G-C could affect the bending of local RNA helices [41]. However, because DA-TLC1 retains the internal loops and bulges throughout these regions, these changes are unlikely to substantially influence the overall persistence length of the arms of DA-TLC1. Therefore, despite lower in vitro activity, DA-TLC1 is likely more biologically relevant than previously reported in vitro in functioning TLC1 variants.

Variants of DA-TLC1 with fewer determined arms may also prove useful for future studies. Promoting lower energy folding of the Est1 and terminal arms together, or just the Ku arm alone, was sufficient to maintain telomeres at near wild-type length in vivo, while also rescuing telomerase activity in vitro at a level similar to DA-TLC1 (Figure 4). Because these RNA variants have fewer nucleotide mutations than DA-TLC1, they could be considered closer to wild-type TLC1 in structure and function. Furthermore, the determined-arm variants could be used strategically to provide a highly functional telomerase RNA while avoiding mutating regions of interest. For example, an RNA with only the “determined” Ku arm could facilitate both in vivo and in vitro studies of the essential Est1-binding arm.

While DA-TLC1 may provide a useful tool, there remain some limitations. Importantly, DA-TLC1 has reduced RNA abundance in vivo (Figure 2B). Since it maintains nearly wild-type-length telomeres despite low RNA levels, DA-TLC1 may actually have higher per-RNA activity than wild-type TLC1 (similar to TSA-T [26]). However, this may impact the relevance of DA-TLC1 to natural wild-type TLC1 function. Further study will be required to fully understand the reasons for the low RNA abundance, and to assess what effects this has on telomerase holoenzyme function. Disrupting the binding of Ku to its site on TLC1 has been shown to decrease RNA abundance by ~50% [19] and this leads to a significant reduction in telomere length. But this reduction in telomere length is not observed with the determined Ku arm in DK-TLC1 (Figure 4). The loss of Sm_7_ binding to TLC1 abolishes the 1157 nt form, leaving only the poly(A)+ species of TLC1 [20,23]; however, none of the alleles in this study show this phenotype (Figure 2C and Figure 4E). Overall, RNA abundance reductions seem to be multifaceted and probably not caused by the lowered holoenzyme subunits’ association with the RNA. Altering the mutations used to create the determined arms may help to mitigate the reduced RNA levels while maintaining a highly functional telomerase RNA.

Another possible limitation of DA-TLC1 concerns the folding predictions for the distal region of the Ku-binding arm. The predicted folding for the DA-TLC1 Ku arm is similar to the Mfold prediction for the wild-type Ku arm (Figure 4A; Appendix A). However, the Ku arm is one of the most poorly determined regions in the Mfold p-num model of wild-type TLC1, so it is not clear whether this structure is truly biologically relevant. The challenge of modeling the full extent of the Ku arm is also reflected by the fact that two different structure models have been proposed for it based on phylogenetic analysis [9,10]. We attempted to drive the folding of the Ku arm into a secondary structure based on the model that used four related *Saccharomyces* species in order to reduce the influence from more divergent species [10]. However, while the resulting “DPhyK” Ku arm allele was able to support greater telomerase function in vitro than TLC1, the RNA had very low RNA abundance and greatly shortened telomeres (Figure 4). These phenotypes were more severe than those observed when deleting the Ku-binding site in TLC1∆48 [19,42], indicating that the RNA has greater defects than the simple loss of Ku heterodimer binding. This may even suggest a previously unknown role of portions of the Ku arm RNA in TLC1 stability. Thus, it is unclear why the secondary-structure-driven Ku arm caused poor telomerase RNA function in cells. It is possible that some of the specific point mutations had unexpected deleterious effects on RNA biogenesis or stability, and/or it may reveal an incomplete understanding of the biologically relevant secondary structure in this region. It is also possible that RNA folding within the wild-type Ku arm is inherently structurally dynamic, and that limiting the secondary structure diversity within the distal region of the determined Ku arm prevented transitions into important conformations. Overall, our attempts to create a fully structurally determined Ku arm demonstrate the complexity of this region of the telomerase RNA and show that further examination of the structural nature of this arm is required. In particular, studying the structure of the Ku-binding arm in its in vivo context, rather than reconstituted or re-folded in vitro, would be valuable. Approaches such as DMS-seq and SHAPE-seq could be useful.

Here, we have reported on a newly designed variant of yeast telomerase RNA that is functional both in vivo and in vitro. Because the overall secondary structure of this determined-arm TLC1 is more similar to the wild-type RNA than previously described in vitro-functioning mutants, DA-TLC1 and its variants may prove to be useful tools for studying the structure-function relationships in the holoenzyme. While the further optimization of the specific mutations may create an even more robust DA-TLC1, we have demonstrated that the in vitro functionality of yeast telomerase RNA can be improved by using structural prediction programs to design the RNA arms to fold into more highly determined, lowest-free-energy structures.

Overall, this study provides biochemical and biological evidence that a *trans*-acting, RNP-forming lncRNA with a largely known secondary structure, that does not fold readily in vitro into its native conformation, can be modified in sequence to promote the biologically relevant secondary structure formation in vitro. This inverse-folding design is achieved here by converting base pairs and unpaired nucleotides to lower the Mfold-predicted free energy of folding, along with the probabilistic folding predictions of the p-num heuristic feature, to promote folding into the biologically relevant active conformation in vitro. This paradigm should prove useful for studying other RNAs and RNPs, provided that the modifications associated with the inverse-folding design do not disrupt sequence-specific binding to key factors or activity of the RNA.

## 4. Materials and Methods

### 4.1. Design of DA-TLC1

DA-TLC1 was engineered by manually introducing a series of nucleotide substitutions in the arms of TLC1 and evaluating these mutant transcripts using Mfold p-num RNA secondary structure prediction software [31,34], with two goals: (1) to drive the lowest-free energy folded state into the phylogenetically predicted structure [10], and (2) to increase the “determinedness” of the three RNA arms, guided by the p-num output. Over 30 inverse-designed variations of TLC1 were examined bioinformatically with this approach, ultimately culminating in the reported DA-TLC1, with more well-determined arms than wild-type TLC1, and a lower predicted initial ∆G (−382 kcal/mol, compared to −321 kcal/mol in TLC1; see Figure 1). DA-TLC1 has 59 nucleotide substitutions over its 1157 nt length; i.e., 9.5% of its nucleotides mutated (see Figure 1). In total, 15 nucleotideswere substituted in the terminal arm, 11 in the Ku arm, and 33 in the Est1 arm (Appendix A). A gene-synthesis fragment for *DA-TLC1* was ordered (*Genewiz/Azenta*) and ligated into a *CEN* vector harboring upstream and downstream sequences from genomic *TLC1*.

The DA-TLC1 variants in Figure 1 and Figure 4 (see sequences in Appendix A) were cloned using restriction enzyme sites in *TLC1*. *DK-TLC1* was created by ligating the *Stu*I to *Nco*I fragment of *DA-TLC1* into *TLC1*, adding the determined Ku arm to wild-type TLC1. *DET-TLC1* was created by ligating the *Stu*I to *Nco*I fragment of *TLC1* into *DA-TLC1*, restoring the wild-type Ku arm sequence to *DA-TLC1*.

### 4.2. Experiments in Yeast

All telomerase RNA alleles were expressed from *TRP1*-marked centromeric (*CEN*) plasmids. These plasmids were transformed into strain TCy43 (*MAT-a ura3-53 lys2-801 ade2-101 trp1-Δ1 his3-Δ200 leu2-Δ1 VR::ADE2-TEL adh4::URA3-TEL tlc1Δ::LEU2 rad52Δ::HIS3 [pTLC1-LYS2-CEN]*) [23]. After shuffling out *TLC1* (*LYS2/CEN*) on solid medium containing α-aminoadipate, colonies were restreaked 10 times (~250 generations; approximately 20 days) on synthetic-compete media lacking tryptophan, with each restreak representing ~25 generations of yeast growth.

### 4.3. Nucleic Acid Blots

Southern blots were performed as previously described [14,19,26]. Briefly, cell pellets for genomic DNA isolation were prepared from cultures set from serially restreaked plates. Genomic DNA was isolated (Gentra Puregene system), and equal amounts were digested with *Xho*I, electrophoresed through a large 1.1% agarose gel at 70 V for 17 h, and transferred to Hybond-N+ Nylon membrane (GE). The blot was then probed for yeast telomeric sequence and a 1627 bp non-telomeric fragment of chromosome IV. Average Y′ telomere fragment length was quantified using the weighted average mobility (WAM) method described previously [19].

Northern blots were performed as previously described [14]. Briefly, total cellular RNA was isolated from yeast cultures using a modified hot-phenol RNA isolation method [43]. After boiling, ~10 μg of total RNA was separated by Urea-PAGE, transferred to a Hybond-N+ Nylon Membrane (GE), UV-crosslinked (Spectrolinker XL-1500 UV Crosslinker, using two runs of the “Optimal Crosslink” setting), and pre-hybridized in Church buffer for 10 min at 55 °C. The membrane was then probed for the 3′ end of TLC1 (nts 906–1140; [26]). Relative abundances were determined by normalized TLC1 levels to U1 snRNA. Since U1 is far more abundant, 100-fold less U1 probe was used relative to TLC1 probe.

RNA dot blots were performed as previously described [26]. These blots are necessary to detect TSA-T RNA abundance since it does not denature in urea-PAGE gels due to its very long dsRNA arms. Total cellular RNA was extracted from late-log or early-stationary phase yeast cultures by a slightly modified version of the “hot phenol” RNA isolation method [43]. After boiling, 5 μg of RNA was spotted twice onto Hybond-N+ Nylon Membrane (GE). The membrane was cut in half (such that the two dots of each sample were separate membrane sections), air-dried, UV-crosslinked (SpectroLinker XL-1500 UV Crosslinker, using two runs of the “Optimal crosslink” setting), and pre-hybridized in Church buffer at 55 °C for 10 min. One membrane was probed for the 3′ region of TLC1 shared by the analyzed alleles (nucleotides 906 to 1140), while the other was probed for the U1 snRNA [44]. Telomerase RNA levels were normalized to U1 levels.

### 4.4. Reconstituted Telomerase Activity Assays

Linearized “run-off” dsDNA templates for T7 transcription of telomerase RNAs were made using PCR (with low-error DNA polymerase) of the gene with a T7 promoter included at the 5′ end of the forward primer. In vitro telomerase activity assays were performed as described previously [14]. Briefly, linear DNA template encoding of each telomerase RNA was mixed with plasmid containing T7-ProA-Est2 in an RRL transcription and translation system. Telomerase RNP was immunopurified with IgG-Sephadex beads and the beads were washed. Bead-bound telomerase was then incubated with a telomeric primer, dNTPs, and [α-^32^P]-dGTP. Products were electrophoresed through a 10% polyacrylamide/TBE/urea gel and imaged using phosphor screens and a Typhoon 9410 Variable Mode Imager. As an internal control for product recovery and loading, ~1 nM [γ-^32^P] end-labeled primer was added before the telomerase reaction. Activity levels were normalized to the internal control.

## Figures and Tables

**Figure 1 ncrna-09-00051-f001:**
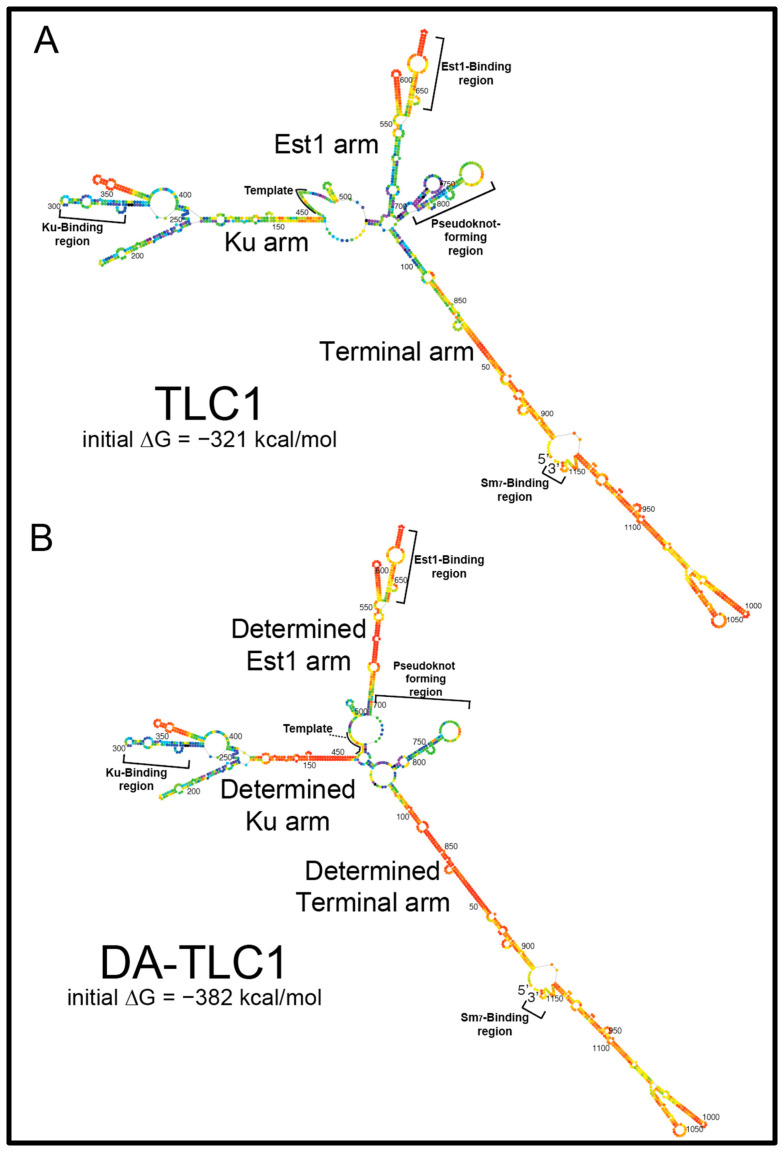
Determined-arm DA-TLC1 RNA: inverse-designed to fold more stably than wild-type TLC1 into the biologically relevant secondary structure. Shown are the lowest-free-energy Mfold p-num computational secondary structure predictions for wild-type and Determined-Arm TLC1 (DA-TLC1). (**A**) Mfold secondary structure model for wild-type TLC1. Nucleotides are colored according to the p-num feature of the Mfold web server, which conveys the “determinedness” of low-free-energy RNA folding predictions, with the most well-determined nucleotides on the red end of the ROYGBIV spectrum [34]. The initial ∆G of folding of wild-type TLC1 was predicted by Mfold to be −321 kcal/mol. (**B**) Lowest-free-energy Mfold secondary model for DA-TLC1, with nucleotides colored in the p-num format as in A. The determined Ku arm was designed to fold into the phylogenetically derived structure for the wild-type arm [10]. The initial ∆G of folding was predicted to be −382 kcal/mol.

**Figure 2 ncrna-09-00051-f002:**
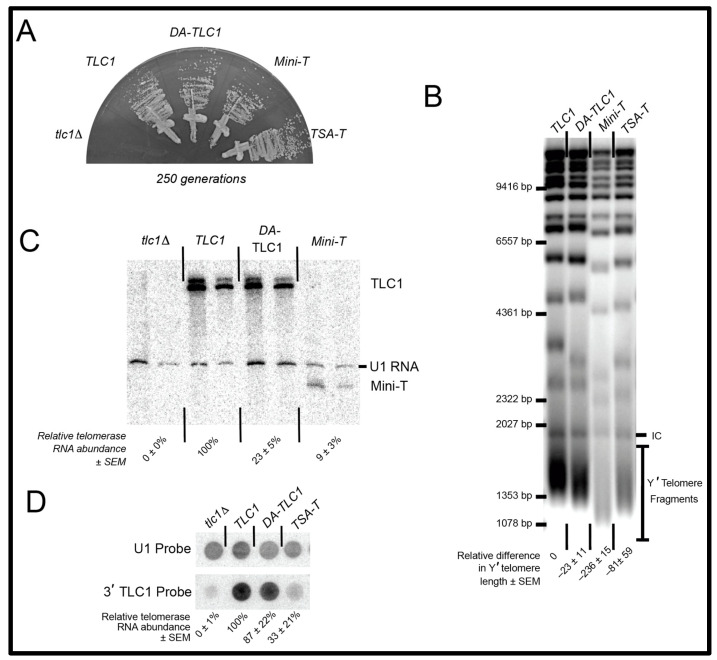
DA-TLC1 RNA can function in place of wild-type TLC1 in vivo, maintaining slightly shorter telomeres and exhibiting reduced abundance. (**A**) Like wild-type TLC1, DA-TLC1 allows yeast cells to grow perpetually and avoid senescing. Each telomerase RNA allele was harbored on a centromere-containing plasmid in a *tlc1*∆ *rad52*∆ yeast strain, and restreaked for over 250 generations. Growth shown at 250 generations (10 restreaks on plates). The *tlc1*∆ strains senesce by ~125 generations. (**B**) Telomeres in DA-TLC1 cells are ~23 bp shorter than wild-type cells. Shown is a Southern blot of *Xho*I-digested genomic DNA probed with telomeric repeat sequences. Change in telomere length relative to wild-type TLC1 is indicated for two isolates, ±S.D. (**C**) The abundance of DA-TLC1 RNA is reduced compared to wild-type. Two independent isolates at 250 generations are shown. Northern blot showing reduced DA-TLC1 RNA levels compared to wild-type. (**D**) Dot blot of total cellular RNA probed for the shared 3′ end of TLC1 (upper panel), which is common to all RNA variants, and normalized to U1 snRNA control spots (lower panel). Normalized telomerase RNA abundance relative to TLC1 from four isolates is indicated, ±standard error. Dot blots are required to study TSA-T since this RNA allele does not denature in urea-PAGE gels used for northern blotting (as in (**C**)) due to its long dsRNA arms.

**Figure 3 ncrna-09-00051-f003:**
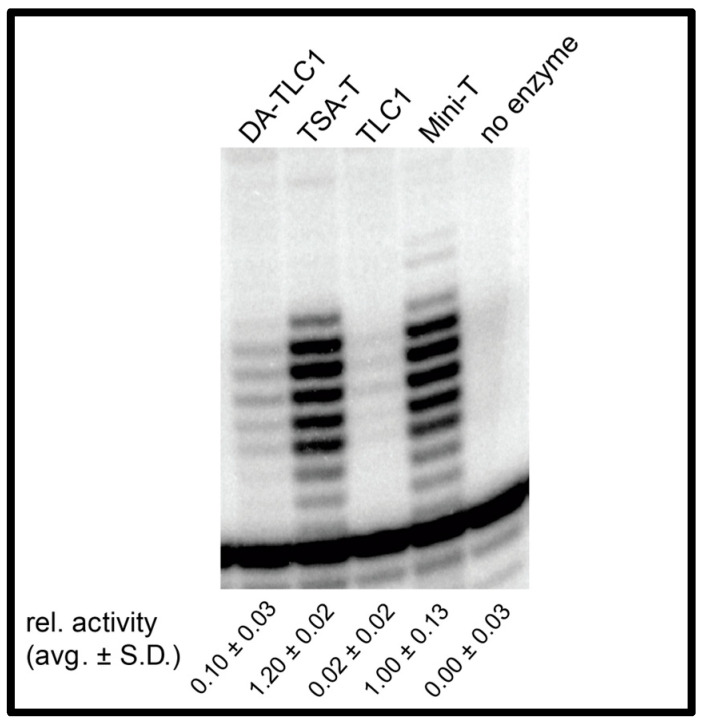
DA-TLC1 telomerase activity is increased compared to wild-type TLC1. The indicated RNAs were synthesized in a rabbit reticulocyte lysate transcript-translation system along with Est2 (TERT), immunopurified and assayed for activity on a telomeric primer in the presence of [α-^32^P]-dGTP. Activity quantification is normalized to internal loading control and is the average ± standard deviation relative to Mini-T activity.

**Figure 4 ncrna-09-00051-f004:**
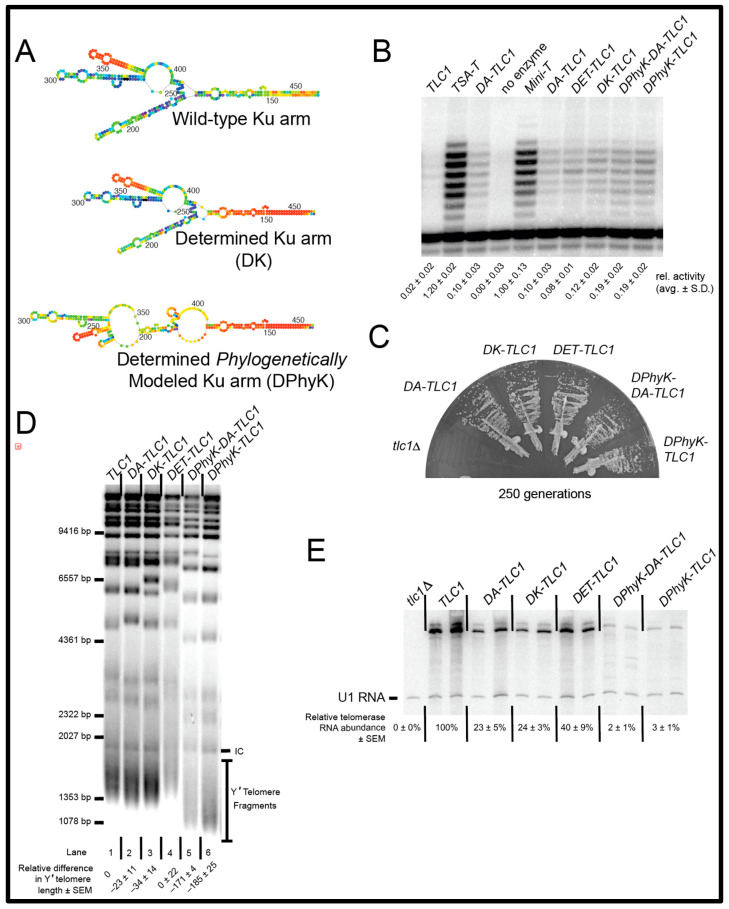
Individual inverse-designed arms each contribute to increased DA-TLC1 activity in vitro and decreased RNA abundance in vivo. (**A**) Mfold p-num models for the Ku arm of wild-type, DK-TLC1, and DPhyK-TLC1 alleles. (**B**) DA-TLC1 supports telomerase function in an in vitro reconstituted telomerase assay. As in Figure 3, TLC1 variants were co-expressed with ProA-Est2 in an in vitro transcription and translation system, co-immunopurified, and reacted with a telomeric DNA primer. Averages ± standard deviation of 2–4 activity assay replicates of each enzyme preparation are shown. (**C**) Cells expressing Determined-Arm TLC1 variants grow well and do not senesce. Telomerase RNA alleles were expressed from *CEN* plasmids in a *tlc1*∆, *rad52*∆ strain. Growth at 250 generations is shown. (**D**) Telomere Southern blot shows that DA-, DK-, and DET-TLC1 alleles support telomeres near wild-type length, whereas DPhyK-TLC1 and -DA-TLC1 exhibit very short telomeres. (**E**) Northern blot shows that, whereas most inverse-designed TLC1 alleles have modestly reduced RNA abundance, DPhyK alleles have very low levels.

## Data Availability

Not applicable.

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
