# Peer review of "Inverse-Folding Design of Yeast Telomerase RNA Increases Activity In Vitro"

_ncrna, 2023, doi:10.3390/ncrna9050051_

Round 1

Reviewer 1 Report

The authors report on newly designed variants of yeast telomerase RNA that are functional both in vivo and in vitro, in particular on DA-TLC1 with predicted higher structural determination of telomerase RNA structure. They compare the newly generated TLC1 forms with previously generated truncated TLC1 variants, as well as with WT TLC1.

Overall, the MS presents interesting results. Indeed, the newly described TLC1 functional variant with the same size as WT is more useful and relevant in structure-functional studies of yeast telomerase RNA. All experiments were performed correctly and results are sound and reliable. Also, presentation, interpretation and discussion of the results is comprehensible and adequate. The quality of the MS is thus high and will be of interest mainly to yeast telomerase researchers.

I would suggest to support the structural free energy-based predictions performed with Mfold by some experimental approach, as e.g., DMS-seq or SHAPE-seq. These results would greatly increase the iompact of the study and demonstrate if the introduced base changes affect only stiffness of the TLC1 secondary structure, or if they also induce any indirect changes in other regions of TLC1. It would also be useful to examine whether Ku-interaction with TLC1 is preserved in the TLC1 variants where the Ku arm is affected.

Minor points:

Lines 188-189: “we examined 188 average telomere length and it to be similar to wild-type cells.“ please, correct the wording.

Lines 410-411: “The challenge of modeling the full extent of the Ku are is also reflected by...“ – correct to “arm“

Reviewer 2 Report

In the  article entitled Inverse-folding design of yeast telomerase RNA increasesactivity in vitro” Kevin J. Lebo and David C. Zappulla used their very rich experience in the study of budding yeast telomerase and in particular Saccharomyces cerevisiae telomerase RNA, TLC1, to design the full-length RNA component of this RNP enzyme which would provide its high enzymatic activity in vitro.

Elegantly using Zuker Mfold software for inverse-folding design they obtained a full-length RNA with 59 mutations in its various parts, the activity in the in vitro system of which was 5 times higher than that of wild-type RNA. Thus, the authors took an important step towards solving the problem of this RNA, as well as corresponding RNP folding in the cell. Considering the revolution in cryo-EM resolution taking place before our very eyes, we can wish the authors to be among the first to be able to obtain the complete spatial structure of telomerase with a resolution close to atomic. To solve many problems associated with the mechanism of this enzyme, as well as the regulation of its activity, the transition from secondary structure schemes which the authors have used so far to a true spatial structure will be extremely important. So far, it has been possible to do this only for small segments of various telomerases, moreover, with insufficiently high resolution. The article is very well written and presented in clear form (by the way, the reviewer had the opportunity to see its full text earlier in the form of a preprint in bioRxiv). It should be published now in a specialized journal.

Round 2

Reviewer 1 Report

Unfortunately, the authors are not willing to perform any of the two suggested experiments to support their results intepretation and conclusions. While in the case of Ku binding, I do not find it essential,, in the case of experimental analysis of secondary structures, this is clearly the missing link in the interprettion of structure-function relationships. The detection of activity shows the final outcome of sequence modifications, but the influence on structure and its stability is only predicted. This considerably weakens this, otherwise high qualty study.